# Biogeographic regionalization by spatial and environmental components: Numerical proposal

**Mayra Flores-Tolentino**[1], **Leonardo Beltrán-Rodríguez**[2], **Jonas Morales-Linares**[1],
**J. Rolando Ramírez Rodríguez**[1], **Guillermo Ibarra-Manríquez**[3], **Óscar Dorado**[4], **José Luis Villaseñor**[5]*

**1** Centro de Investigación en Biodiversidad y Conservación, Universidad Autónoma del Estado de Morelos, Cuernavaca, Morelos, Mexico, **2** Jardín Botánico–Instituto de Biología, Universidad Nacional Autónoma de México, Coyoacán, Ciudad de México, Mexico, **3** Instituto de Investigaciones en Ecosistemas y Sustentabilidad, Universidad Nacional Autónoma de México, Morelia, Michoacán, Mexico, **4** Centro de Educación Ambiental e Investigación Sierra de Huautla, Universidad Autónoma del Estado de Morelos, Cuernavaca, Morelos, Mexico, **5** Departamento de Botánica, Instituto de Biología, Universidad Nacional Autónoma de México, Ciudad de México, Mexico

* vrios@ib.unam.mx

**Data Availability Statement:** All relevant data are within the manuscript and its Supporting Information files.

## Abstract

Regionalization through the analysis of species groups offers important advantages in conservation biology, compared to the single taxon approach in areas of high species richness. We use a systematic framework for biogeographic regionalization at a regional scale based on species turnover and environmental drivers (climate variables and soil properties) mainly of herbaceous plant species richness. To identify phytogeographic regions in the Balsas Depression (BD), we use Asteraceae species, a family widely distributed in Seasonally Dry Tropical Forest (SDTF) and the most diverse of the vascular plants in Mexico. Occurrence records of 571 species were used to apply a quantitative analysis based on the species turnover, the rate of changes in their composition between sites ($\beta$-Simpson index) and the analysis of the identified environmental drivers. Also, the environmental predictors that influence species richness in the SDTF were determined with a redundancy analysis. We identified and named two phytogeographic districts within the SDTF of the BD (Upper Balsas and Lower Balsas). According to the multi-response permutation procedure, floristic composition of the two districts differs significantly, and the richness of exclusive species in Upper Balsas was higher (292 species) than in the Lower Balsas (32 species). The proportion of Mg and Ca in the soil and the precipitation of the driest three-month period were the environmental factors with greatest positive influence on species richness. The division of geographic districts subordinated to the province level, based on diverse families such as Asteraceae, proved to be appropriate to set up strategies for the conservation of the regional flora, since at this scale, variation in species richness is more evident. Our findings are consistent with a growing body of biogeographic literature that indicates that the identification of smaller biotic districts is more efficient for the conservation of biodiversity, particularly of endemic or rare plants, whose distribution responds more to microhabitats variation.

**Funding:** The Consejo Nacional de Ciencia y Tecnología (CONACYT) provided financial support to MFT (scholarship 732218) to carry out her doctoral studies.

**Competing interests:** The authors have declared that no competing interests exist.

## Introduction

The geographical distribution of biodiversity shows patterns that repeat in different taxa [1, 2]. These biogeographic patterns allow the recognition of biotic components, defined as sets of spatio-temporally integrated taxa due to a common history, which characterize geographic areas or biogeographic regions [2–4]. A biogeographic regionalization is a hierarchical system that classifies geographic areas in terms of their endemic biota [2, 5, 6], allowing the definition of homogeneous regions generated from sets of species and the identification of factors that potentially influence their distribution [7]. Biogeographic regionalization is essential to understand the spatial distribution of biodiversity [8], as well as to identify important areas for their richness of species and endemisms, which allow to propose strategies for their conservation [9, 10]. Consequently, sets of species with the same distribution are the ideal model to recognize biotic components, biogeographic regions, and provinces [11, 12].

At present, the availability of databases such as the Global Biodiversity Information Facility (GBIF) or the National System of Information on Biodiversity (SNIB-CONABIO), has contributed to improving both our understanding of the distribution of species and the analyzes that allow classifying biogeographic patterns [13, 14]. These databases also allow the application of other methods focused on evaluating species turnover, an equally important component of biogeographic regionalization [6, 15]. Measures of similarity and differentiation of especies are essential tools to assess the effects of isolation by distance or geographic barriers, and to describe changes in species composition along environmental gradients [16]. Regionalization derived from quantitative methods can result in the division of biogeographic districts that other stakeholders can evaluate and replicate [17].

Regionalization through the analysis of species groups offers important advantages in conservation biology, compared to the single taxon approach, especially in areas rich in species, such as tropical dry seasonal forests (SDTF) [18–22]. In these forests, the conservation of threatened bioregions is more successful when the remaining fragments are protected rather than individual species [19, 23]. In this sense, bioregions may act as Biodiversity Hotspots, a concept based on species richness, endemicity and threat [24, 25].

In Mexico, several studies address biogeographic regionalizations using different groups of species (e.g., [26–29]). Despite the interest in regionalization at global scales [13], little is known about regionalization at the provincial level or even at the district or sector level (e.g., [30]). Recently, Morrone [31] in a Mexico's regionalization analysis recognized two regions (Nearctic and Neotropical) and 14 provinces, which allows a general perspective of how different species have assembled in the different geological and climatic conditions. However, biogeographic regionalization, at levels lower than regions or provinces, using groups of representative species, should be more efficient for the application of conservation strategies [32].

The Balsas Depression (BD), in central western Mexico, is one of the provinces characterized by the dominance of the seasonally dry tropical forest (SDTF; 65%) and constitutes a center of diversification and endemism, as well as the biogeographic transition between the Neotropical and Nearctic regions [11, 28]. The complex environmental and biogeographic history of the SDTF conceives it as a heterogeneous biome and difficult to circumscribe [33]. In México, the SDTF is distributed mainly in the Pacific slope from southern Sonora and southwestern Chihuahua to Chiapas and on the gulf slope from Tamaulipas to the Yucatán Peninsula [34]. Different studies carried out in the SDTFs at local scales, have shown that the patterns of plant species diversity and richness are driven by the water availability and the soil properties [35–38]. However, currently few studies (e.g., [39–42]) have focused on the study of the richness' drivers of the SDTF at regional or global scales.

An ideal group for regionalization studies in Mexico is the Asteraceae family, worldwide recognized for its high species diversity [43] and found also among more diverse families in the neotropical SDTF [43]. In Mexico, it is among the most diverse and comprehensively studied families of Angiosperms [44] with 3,057 species [45]. In addition, its species show a significant correlation with the total floristic richness. Therefore, it can be considered as a good biodiversity' surrogate in Mexico [46]. These characteristics also place it as a good surrogate for defining biogeographic subregions in areas poorly explored floristically, such as the SDTF in the BD.

Considering that in the BD the most representative biome is the SDTF, in which the Asteraceae are widely distributed, our objectives were: 1) to determine a biogeographic regionalization of the SDTF in the BD, based on the Asteraceae' species turnover, 2) identify the environmental predictors that determine the Asteraceae' species richness in the SDTF and, 3) analyze the relationship between turnover species patterns with environmental predictors. It is known that the changes in the environmental conditions of each region explain the patterns of species turnover [47]. Therefore, we hypothesize that an environmental differentiation will occur in the SDTF of the BD, which will cause the species turnover of the Asteraceae and will allow us to identify biogeographic regions. The regionalization in the BD will make it possible to understand the distribution patterns of the Asteraceae, improve the understanding of their spatial distribution and identify areas with greater relevance due to their species richness, this information will be useful for future conservation studies.

## Materials and methods

### Study area

The BD is one of the 17 provinces proposed by Rzedowski [28], located in central Mexico, with an area of 115,005 km$^2$; it includes part of the states of Guerrero, Jalisco, Mexico, Michoacán, Morelos, Puebla, and Oaxaca. The BD stands out for its species richness and endemism, the flora comprises 4,442 to 6,800 species of vascular plants, of which 337 are endemic [28, 48, 49]. The biome characteristic in the province is the SDTF [11], with a surface area of 74,753 km$^2$ (65% of the total surface of BD). In Mexico, the SDTF is considered one of the most distinctive and diverse biomes with more than 6,000 species of plants, 45% endemic [34, 50].

### Taxonomic study group

The Asteraceae family stands out worldwide for its species richness; with more than 23,000 species, ranks among the most diverse of flowering plants [43]. In Mexico, Asteraceae is found in practically all terrestrial ecosystems, which is due to its great species richness and its wide range of altitudinal distribution (from sea level to high mountain moorlands). Most of the Asteraceae species are herbaceous, and this life form is the richest in species in the SDTF [51]. However, most of the ecological studies in SDTF have focused on tree species [41, 52]. Therefore, evaluating the herbaceous life form would provide new information on the environmental factors that drive species richness and plant composition in the SDTF. This bias must be eliminated since herbs constitute the growth form with the highest species richness in this biome [51].

### Spatial data

All records of the Asteraceae family reported for the BD were extracted from the SNIB-REMIB and MEXU-UNIBIO databases. A total of 60,005 records were obtained from this search, which were systematically cured following the recommendations of Castillo et al. [53] and

Chapman [54]: as 1) the records that did not have coordinates were georeferenced in Google Earth (https://www.google.com/earth/), using locality name and description of the herbarium specimen, 2) exclude the records that were outside the limits of the BD, and 3) eliminate the records that could not be georeferenced. We reviewed and corrected spatial errors, such as the coordinates of erroneously georeferenced locations, using the ArcGis 10.2 program [55]. After the curatorial evaluation, the BD final database consisted of 21,501 Asteraceae records, corresponding to 789 species. From these records, only 7,479 belong to the tropical portion or SDTF and the others to the temperate zone; they record 571 species, of which 15% are trees, 27% shrubs, and 58% herbs.

## Spatial analysis

The process for the biogeographic regionalization of the SDTF of the BD consisted of a series of analyzes that are detailed in the following sections. Fig 1 shows the workflow for the different analyzes carried out that resulted in regionalization and the relationship of the groups identified with environmental predictors.

## Cluster analysis

With the use of the Biodiverse v.2.1 program [56], we identified floristic districts within the tropical portion of the BD [56]. This program is a tool for the spatial analysis of diversity that uses indices based on taxonomic relationships. The refined database, including the geographic coordinates and the taxonomic identification of each record, registered in a set of grid-cells of $0.25° \times 0.25°$ size was imported into Biodiverse.

We calculated a species turnover matrix for all cell pair combinations, using the *β-Simpson (βSim)* dissimilarity index [57]. This index reduces the effect of the species richness imbalance among the grid-cells, calculated through the following expression:

$$\beta Sim_{i,j} = 1 - \frac{a}{a + \min{(b, c)}}$$

Where $a$ is the number of common species shared in cells $i$ and $j$, $b$ is the number found in $i$ but not in $j$, and $c$ is the number found in $j$ but not in $i$. A value close to 0 for $\beta Sim$ indicates that high proportion of taxa are shared (low turnover), while a high value (>0.8) means a low proportion of shared taxa (high turnover) between two cells.

Grid cells containing fewer than five records were excluded from the analysis, as small sample sizes can potentially cause considerable distortions in dissimilarity analyzes [58, 59]. We integrated the data from the excluded grid cells into their neighboring ones; these exclusion criteria reduced the number form 159 (original subdivision) to only 122 grid cells (Fig 2).

The dissimilarity matrix was used (*βSim*) for cluster analysis, using WPGMA clustering method (weighted pairing groups method using arithmetic mean) by means of the Biodiverse program. Results of cluster analysis made it possible to identify groups of cells with sets of similar species, used to subdivide the SDTF in the BD. The WPGMA algorithm evaluates the contributions of the clusters by the number of terminal nodes (grid cells of the data set) they contain, ensuring that each cell contributes equally to each fused group of which it is part [60].

We reassigned the unrepresented grid cells to those groups with higher representation. We evaluated statistically the resulting groups by the Multi-response Permutation Procedure (MRPP) analysis [61]. This analysis allowed determining if the floristic composition of the regions differed significantly within the SDTF.

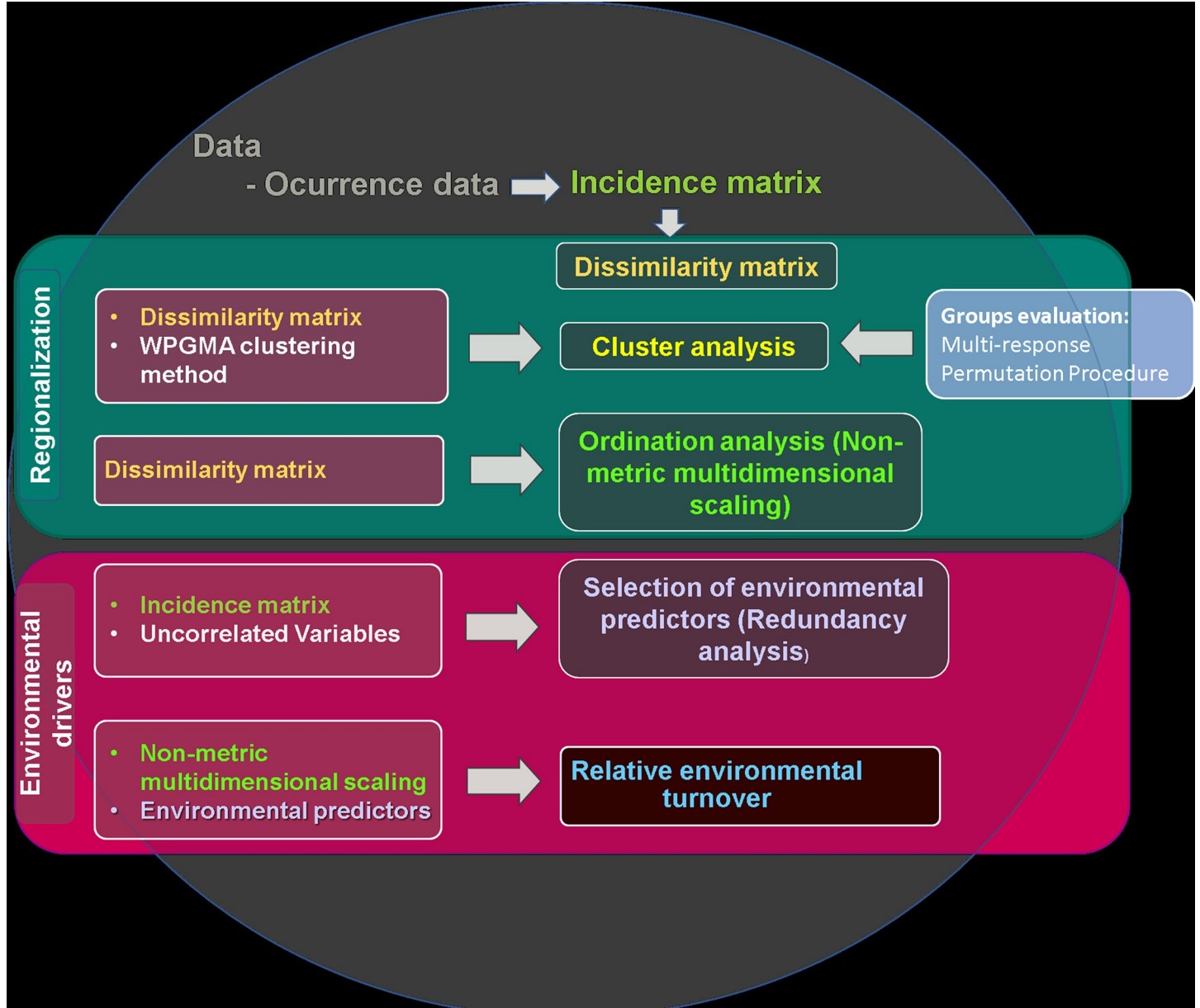

**Fig 1. Schematic workflow of the proposed framework for biogeographic regionalization and spatial analysis at the regional level.** Each panel shows the analysis carried out and the inputs used.

### Ordination analysis

Ordination using non-metric multidimensional scaling (NMDS) is a widely used technique to obtain low-dimensional projections of multivariate data, by organizing objects (in this case, a set of grid cells) along the reduced axes according to the taxonomic composition [60]. We carried out the NMDS analysis using the 'metaMDS' function from the Vegan package in R statistical software. Pairwise distances were calculated using $\beta Sim$. Among the statistics provided by the analysis is a stress value, which reflects the amount of error in the correlation between pairwise distances in the original matrix and a matrix calculated with the NMDS. Stress values of $\leq 0.1$ indicate excellent representation in reduced dimensions, $\leq 0.2$ good and values $\geq 0.3$

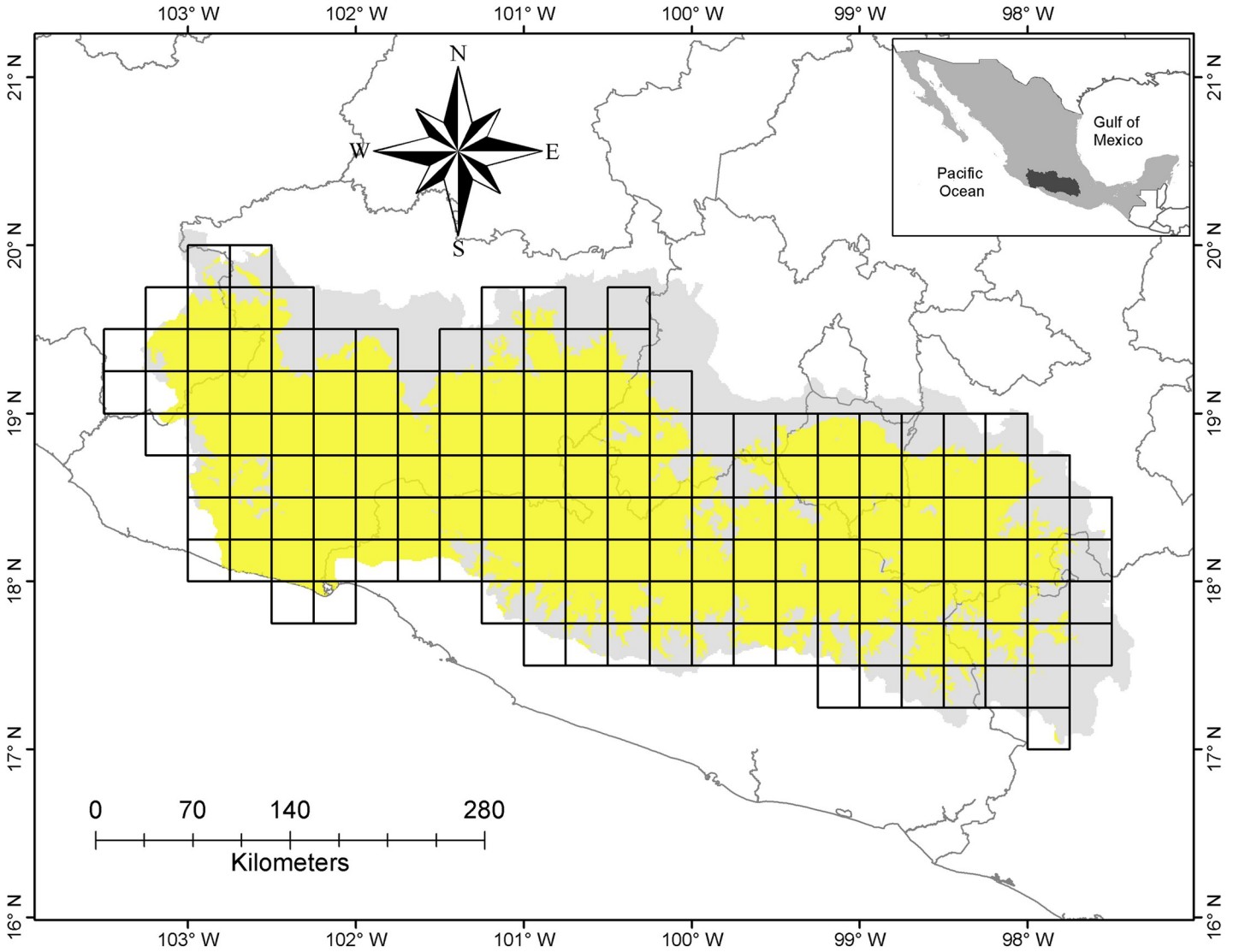

**Fig 2. Location of the Floristic Province of Balsas Depression in Mexico (dark grey area).** Distribution of seasonally dry tropical forest (yellow area) in this floristic province, divided in squares of 0.25˚ × 0.25˚ arc-min.

provide a poor representation [62]. We extracted and projected on a map in ArcGIS the values of each cell of the first and second axis of the NMDS.

## Selection of SDTF environmental predictors

First, we considered a set of 58 environmental variables at a resolution of 1 km$^2$: 26 climatic [63], 9 edaphic, 9 topographic, and 14 that include remote sensing data [64]. Subsequently, we performed a Pearson correlation analysis to rule out variables with high collinearity values. Once selected the uncorrelated variables, we extracted the values of each 1 km$^2$ pixel using ArcGis 10.2. These environmental values were added to a 0.25˚ × 0.25˚ grid cell (122 cells in total), using the average values of each cell.

We identified the environmental predictors with the highest explanatory value of the species richness of the SDTF in the BD. This method allows extracting and summarizing the

variation in a set of response variables that can account the set of explanatory variables [65]. For this analysis, we used both an incidence matrix of 571 Asteraceae species and another with environmental data of 32 uncorrelated variables (S1 Table). The data were standardized to z-values, based on the mean and standard deviation [66], which is used to standardize values to the same scale. We performed a Redundancy analysis (RDA) using the "rda" function of the Vegan package [67] in the statistical software R 3.6.3 [68]. Finally, we selected the most parsimonious model and the variables with the greatest significance ($p$ <0.001, 999 permutations).

**Relative environmental turnover.** To calculate the relationship between environmental predictors and species turnover, we applied the relative environmental turnover (RET) method. For this, we adjusted the NMDS results with the matrix of previously selected environmental predictors, using the vector adjustment of the *envfit* function of the Vegan package in the statistical software R. The significantly related environmental predictors to the turnover patterns ($p$ <0.001, 999 permutations) were shown as vectors in the NMDS plot.

## Results

### Cluster analysis

Although the clustering identified eight groups in the BD (Fig 3A), two are the main floristic groups considering the number of squares that encompassed, named Upper Balsas and Lower Balsas (groups three and four, respectively). The spatial patterns of the species characterizing each group showed a significant correlation between them. The Lower Balsas had a greater dissimilarity in its species composition, allowing recognition of other four poorly differentiated groups (groups 5–8, Fig 3B). The differentiated groups shown in the dendrogram (Fig 3A) are represented by species exclusive to these groups (S2 Table).

According to the results of the MRPP, the floristic composition was statistically different ($p$ <0.001) between the two consensuses, which from now on we will refer to as Upper Balsas and Lower Balsas districts or biogeographic districts (Fig 4). The exclusivity of the species within the districts is greater in the first ($\delta$ = 16.66, N = 292 restricted species) than in the last one ($\delta$ = 11.75, N = 32 restricted species).

The biogeographic tracks (collecting points linked by a minimum spanning tree) of the exclusive species of each biogeographic districts support the subdivision obtained by the classification methods (Fig 4). Each identified biogeographic districts meets environmental and orographic conditions that have allowed the differentiation in its species composition. For example, the species exclusive to the Lower Balsas district (western biogeographic track; Fig 4) show a preference for geographical areas at lower altitude (<750 m). The opposite situation occurs with the species that make up the eastern biogeographic track in the Upper Balsas district, because these species prefer higher altitudes (>750 m).

### Ordination analysis

The NMDS analysis provides two dimensions, where the first axis (NMDS1; Fig 5) indicates a geographic break that differentiates the BD in two geographic areas (Fig 5A); both areas coincide relatively well with the pattern obtained in the classification method. The second axis (NMDS2) shows an abrupt turnover in the Lower part of BD (Fig 5B), distinguishing a different area at the east-central part.

### SDTF environmental predictors

The redundancy analysis allowed selecting the most important variables that influence the Asteraceae species richness in BD. The most parsimonious model provided nine variables that

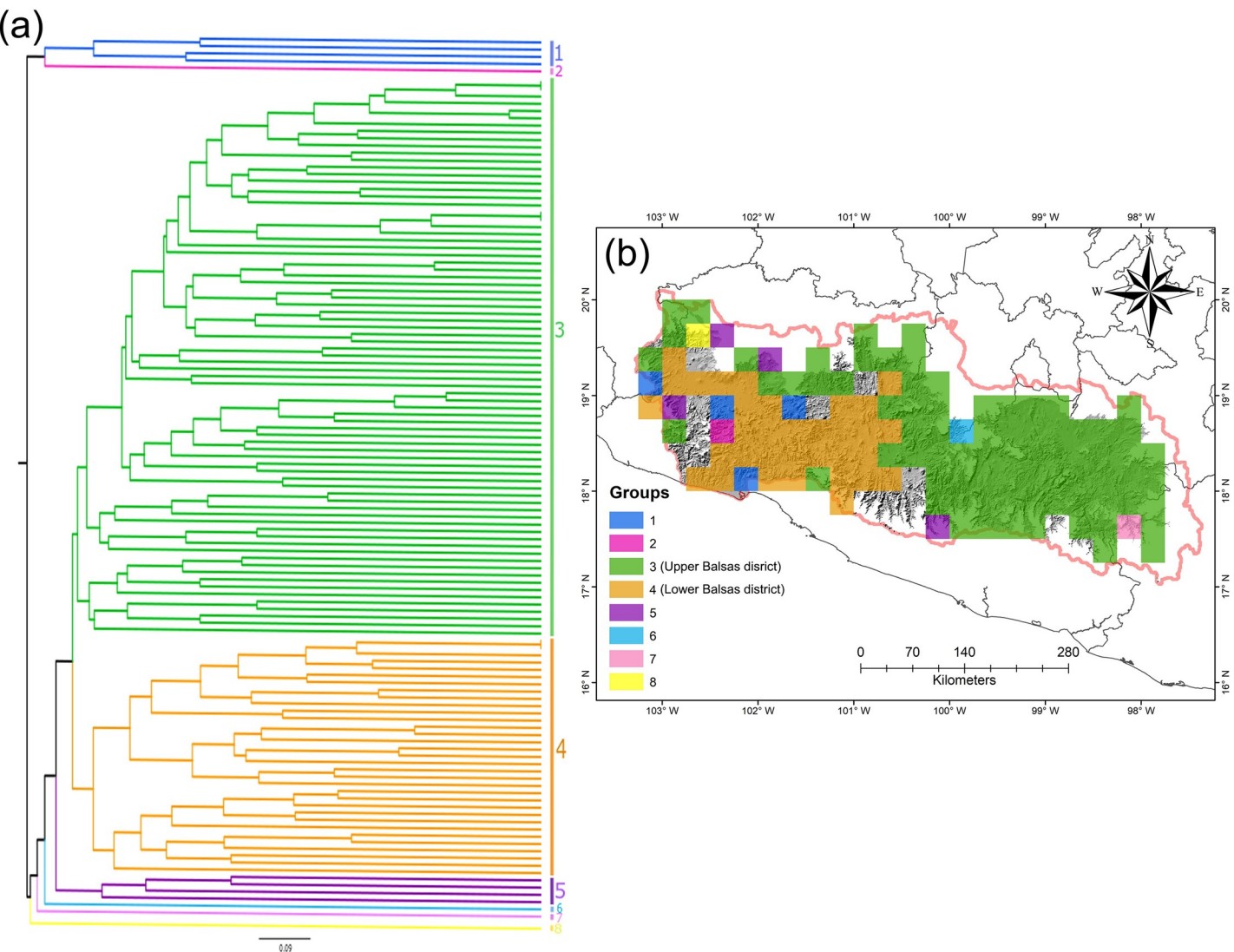

**Fig 3. Cluster analysis (*β-Simpson* dissimilarity coefficient) showing the floristic dissimilarity of the grid squares with the Asteraceae species from the seasonally dry tropical forest in the Balsas Depression, Mexico.** (a) Dendrogram showing floristic dissimilarity. (b) Balsas Depression where the colors correspond to the groups shown in the dendrogram.

explained 39.65% ($p = 0.05$) of total accumulated variance, while the combination of the variables with greatest significance explained 28.01% ($p = 0.001$).

## Relative environmental turnover (RET)

The RET analyses suggests an acceptable fit of the environmental data, with a stress value of 1.18, in relation to the species turnover in the NMDS' ordination (Fig 6). The results suggest that precipitation availability and soil properties (Mg and Ca nutrients) play an important role in the Asteraceae richness of SDTF in the BD (Table 1). The species composition of each district was influenced by the availability of Ca and Mg in the soil. The most diverse district (Upper Balsas) registered a higher Ca concentration (mean 0.93 mg, sd ± 0.49) than the Lower Balsas (0.40 mg ± 0.16). In contrast, Mg is slightly higher in the Lower Balsas (0.32 mg ± 0.07) than in the Upper (0.29 mg ± 0.08).

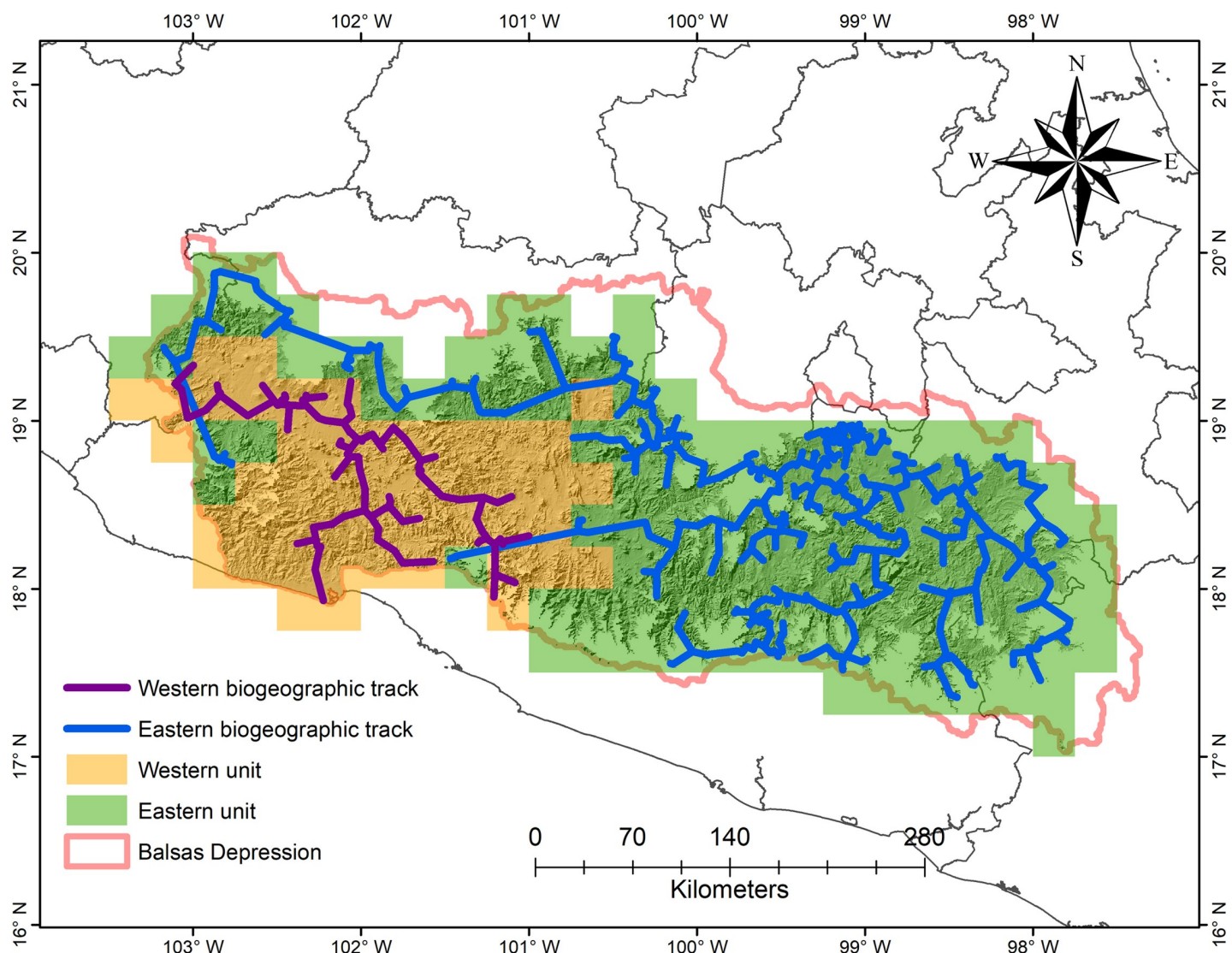

**Fig 4. Phytogeographic subdivision of the seasonally dry tropical forest in the Balsas Depression, Mexico.** The purple biogeographic track links by means of a minimum spanning tree the collecting points of the species exclusive to the Lower Balsas and the blue line those of the Upper Balsas.

## Discussion

Our results agree with previous biogeographic studies developed in the BD, using the *Bursera* (Burseraceae) trees [28, 41, 52], which recognize two districts. The difference with these studies, except for Gámez et al. [41], is that they do not provide a geographic delimitation that circumscribes these two phytogeographic districts. Gámez et al. [41] identified three areas of endemism for *Bursera*, two of them including part of BD (*sensu* [69]): i) the Balsas Occidental and ii) the Balsas Oriental-Tehuacán /Cuicatlán-Tehuantepec. Despite the discrepancy in the geographic boundaries and the names of the districts with the work of Gámez et al. [41], the district located in the East of the BD, is the region with the highest number of species.

Some studies have shown that precipitation and soil properties affect current patterns of species diversity in the tropical dry forest (e.g., [35, 70, 71]); in this sense, our results also indicate that precipitation seasonality is the most important variable for explaining species

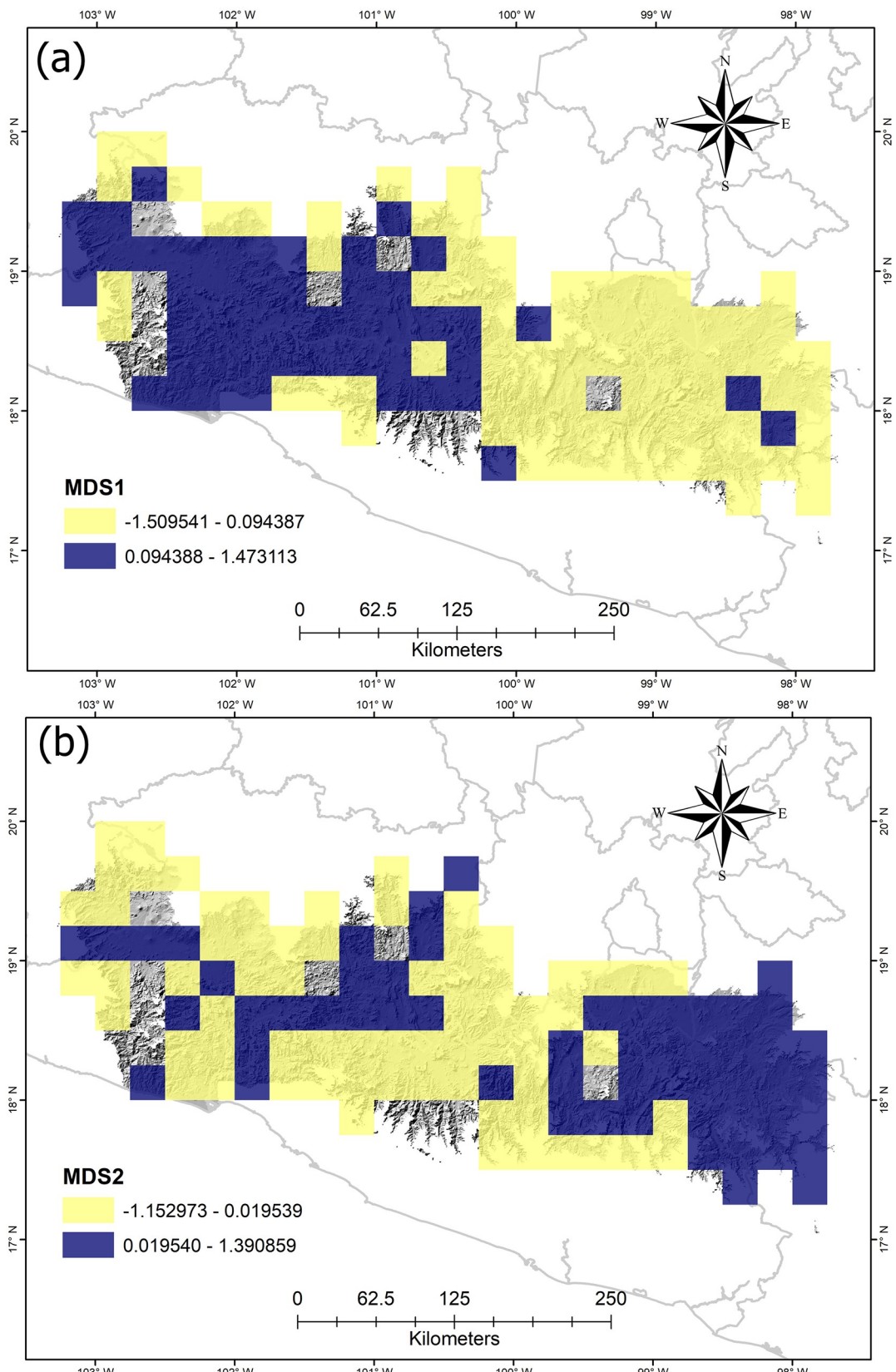

**Fig 5.** Asteraceae species turnover measured with the non-metric multidimensional scaling method (NMDS) for (a) axis 1 (NMDS1) and (b) axis 2 (NMDS2). The colors mark the two turnover ordering classes.

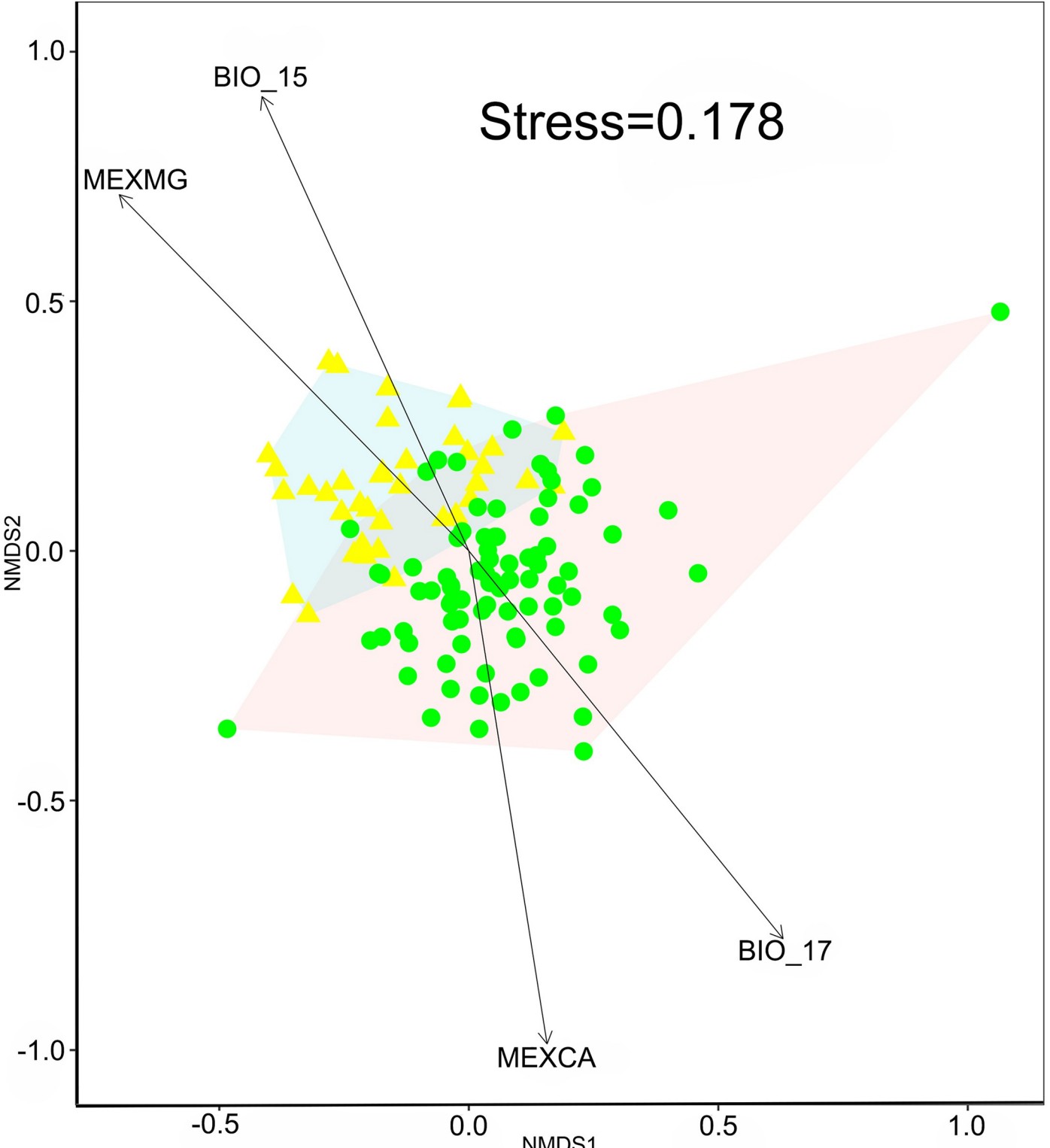

**Fig 6. NMDS ordination and environmental predictors (vectors) as predictors of environmental turnover, calculated for 122 grid cells, distributed along the Balsas Depression, Mexico.** The vectors shown include only the variables with a significant effect ($p < 0.001$) on the NMDS ranking. BIO_15: Precipitation Seasonality (coefficient of variation in %); BIO_17: Precipitation of the driest four-month period; MEXMG: Magnesium content; MEXCA: Calcium content. The circles correspond to the grid cells of the Upper Balsas and the triangles to the cells of the Lower Balsas.

**Table 1. Variables that constitute the most parsimonious model of redundancy analysis.**

|          | Df | AIC    | F          |
|----------|----|--------|------------|
| MEXMG    | 1  | 621.35 | 9.8698**   |
| MEXCA    | 1  | 620.48 | 8.9947**   |
| BIO_17   | 1  | 614.96 | 3.6486**   |
| BIO_15   | 1  | 614.32 | 3.0444**   |
| BIO_02   | 1  | 613.7  | 2.4598*    |
| MEXPH    | 1  | 613.69 | 2.4462*    |
| MODISDIC | 1  | 613.66 | 2.4148*    |
| EVAANUAL | 1  | 613.21 | 2.0007.    |
| MEXDEM   | 1  | 612.76 | 1.5746.    |

MEXMG: Magnesium, MEXCA: Calcium, BIO_17: Precipitation of driest quarter, BIO_15: Precipitation seasonality, BIO_02: Mean diurnal range, MEXPH: pH, MODISDIC: Normalized vegetation index December, EVANUAL: Annual real evapotranspiration, MEXDEM: Elevation digital model.

richness in the SDTF. Therefore, the highest Asteraceae richness values concentrate in relatively high and stable humidity conditions, such as those found in the Upper Balsas district. The precipitation of the driest quarter showed a negative correlation with the Asteraceae species richness, suggesting that precipitation stability in the driest months is an important factor determining species richness. These results are similar with those found by Zhang et al. [72], who found a positive correlation between rainfall and the richness of woody plant species in China.

The SDTF plants are subject to a marked rainfall seasonality that varies between years and imposes an important abiotic restriction for secondary stem growth and phenology, especially in the arboreal component [73]. In the case of Asteraceae, the effect of the precipitation seasonality could also be of great relevance; 58.5% of the SDTF Asteraceae species are herbaceous, thus the rainy season must regulate several aspects of their life cycle, for example reproductive phenology [74, 75]. In SDTF, precipitation pulses trigger the biological cycle of many herbaceous taxa, especially the annual species that germinate and reproduce in short periods in synchronization with the climatic patterns [76, 77].

At a global level, different studies carried out in the Neotropics highlight the importance of precipitation in the SDTF's dynamics (e.g., [20, 78, 79]). In Mexico, the studies focused on evaluating the effect of precipitation on the distribution patterns of SDTF species at regional scales [39–42, 70], have also highlighted its importance, results that coincide with what was found in this study.

Some eco-physiological traits of Asteraceae species, such as the development of underground water storage systems, are related to the appearance of secretory tissues efficient in maintaining individuals during droughts. For example, *Ageratina adenophora* develops rhizomes that allow to store water, while *Pittocaulon praecox* and *Roldana lobata*, show abscission of the leaves during the driest season and the accumulation of mucilage and perennial structures that allow regrowth [80]. In this way, the combination of mesomorphic foliar traits and vegetative propagation provide resistance to extreme climatic variation [80, 81], as occurs in the SDTF [77].

It has been observed that most of the Asteraceae species, for example some members of the Eupatorieae tribe forming part of group three (Fig 3), especially distributed in the BD's eastern portion, show a high growth rate, due to its ability to absorb nutrients [80]. This attribute gives them a competitive advantage [80, 82], but there is no information about the fuctional

strategies of Asteraceae species in tropical-dry environments. Nevertheless, the approaches made for other taxonomic groups with predominantly arboreal growth forms [76, 83] may be useful to explain the patterns observed in the species members of group 4 whose distribution is restricted to Lower Balsas. These Asteraceae species have developed mechanisms for survival to drought that may include deep rooting, loss of leaves during the dry season or face this last unfavorable season for their survival in the seed bank.

Another relevant factor accounting for the spatial distribution of Asteraceae species richness of the SDTF in BD were the soil components, although their importance was less than of precipitation. However, it has been documented that the abundance and different functional aspects of the SDTF species correlate with the chemical composition of soil [37, 38]. Werden et al. [38] found that distribution of 94% of the tree species in the SDTF of Costa Rica responds to the chemical characteristics of the soil. Richness and diversity of rare species in warmer soils of tropical forests in Hainan Island, China, correlate significatively with Ca and Mg content [84]. Therefore, in addition to the precipitation regime, Ca and Mg in the soil should influence the floristic differentiation of the Asteraceae family in BD, which is represented mainly by herbaceous species (58%) that are typical indicators of these elements [85]. In summary, there seems to be some correlation between the SDTF phytogeographic areas, and some soil properties, especially at the Upper Balsas, which concentrates the higher proportion of species.

Previous research suggests that other soil components, such as P, Cu, N, and Al, also contribute significantly to soil fertility in the SDTF of Neotropics [34, 36, 37, 86]. However, in our results these elements were not relevant to explain the Asteraceae species richness. One possible explanation lies in the study group (herbaceous versus trees), since nutrients as P and N are known to be key elements for the growth and reproduction of many tropical trees [38, 84], but in high concentrations they can inhibit these physiological functions, especially in species with herbaceous growth form [87].

Both NMDS and clustering analyses proved to be efficient tools to identify floristic assemblages of the SDTF in BD. The analyzes carried out in this study support the hypothesis that species turnover patterns are driven by changes in environmental conditions [47] and that the mechanisms causing the dissimilarity pattern may differ between biogeographic districts. In this research, each biogeographic district showed both climatic (precipitation) and edaphic characteristics, which can explain the differentiation in species composition. In particular, the Lower Balsas shows greater climatic variation (temperature) than the Upper Balsas, which is more stable.

This study applied quantitative and correlative methods that increasingly provide better guides to identify the geographic limits of areas that combine different assemblages of species of the Asteraceae family in the BD. On the other hand, the relevance of this contribution lies in the fact that the applied methods can be replicable with other groups of species and biogeographic regions. In this way, future studies will be able to integrate various groups of biological interest, to know in a more comprehensive way their influence on the formation of phytogrographic regions.

The SDTF is one of the most important biomes due to its high degree of endemism, but also the one most threatened by human activities such as land use change and climate change [88, 89]. Therefore, this approach can be the starting point for the analysis of the effect of environmental predictors on the species, such as the soils of biogeographic districts.

## Conclusion

The use of environmental predictors and representative taxa of biodiversity improves the definition of biogeographic regions. Both the classification and ordination methods used for

regionalization within the BD coincide in the identification of two different floristic district (Upper Balsas and Lower Balsas). On the other hand, the SDTF climatic variation influences the grouping of species and promotes the high diversity of Asteraceae species of the SDTF in the BD. Mapping the geographic patterns of species richness and identifying the relationship between richness and environmental factors is essential to help conserve biodiversity in highly threatened and highly species-diverse environments, such as SDTF. The species richness partitioning into smaller biogeographic districts will allow planning more efficient conservation strategies, for example, focusing on those areas with greater species richness or endemism. Finally, this approach to the study of the spatial patterns that use plants with different growth forms are complementary and probably reflect different evolutionary processes and ecological relationships that have not been fully explored.

## Supporting information

**S1 Table. Variables used for the selection of environmental predictors in the seasonally dry tropical forest of the Balsas Depression, Mexico.**
(DOCX)

**S2 Table. Characteristic species of phytogeographic groupings of Fig 3.**
(DOCX)

## Acknowledgments

CONABIO and the Instituto de Biología, UNAM, are grateful for access to the information stored in the SNIB-REMIB and UNIBIO databases, respectively, which formed a fundamental part of the analysis presented here. M.F.T. is grateful to the Doctorado en Ciencias Naturales of the Universidad Autónoma del Estado de Morelos and Consejo Nacional de Ciencia y Tecnología (CONACYT) for the scholarship support to carry out her doctoral studies. This paper is a product of M.F.T.´s PhD degree at the Doctorado en Ciencias Naturales of the Universidad Autónoma del Estado de Morelos, Mexico. We all are also grateful to Enrique Ortiz for his advice throughout this project and to the work team of cubicle A218 of the Institute of Biology of UNAM for their great ideas that always enrich the discussion sessions.

## Author Contributions

**Conceptualization:** Mayra Flores-Tolentino, José Luis Villaseñor.

**Formal analysis:** Mayra Flores-Tolentino, Leonardo Beltrán-Rodríguez, Jonas Morales-Linares, José Luis Villaseñor.

**Funding acquisition:** J. Rolando Ramírez Rodríguez.

**Investigation:** Mayra Flores-Tolentino, José Luis Villaseñor.

**Methodology:** Mayra Flores-Tolentino, José Luis Villaseñor.

**Project administration:** Mayra Flores-Tolentino, José Luis Villaseñor.

**Supervision:** Mayra Flores-Tolentino, Leonardo Beltrán-Rodríguez, J. Rolando Ramírez Rodríguez, Guillermo Ibarra-Manríquez.

**Visualization:** Mayra Flores-Tolentino, José Luis Villaseñor.

**Writing – original draft:** Mayra Flores-Tolentino.

**Writing – review & editing:** Mayra Flores-Tolentino, Leonardo Beltrán-Rodríguez, Jonas Morales-Linares, J. Rolando Ramírez Rodríguez, Guillermo Ibarra-Manríquez, Óscar Dorado, José Luis Villaseñor.

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
