## [Decision Letter · Decision Letter 0]

7 Apr 2021

PONE-D-21-01262

Biogeographic regionalization by spatial and environmental components: a numerical proposal

PLOS ONE

Dear Dr. Flores-Tolentino,

Thank you for submitting your manuscript to PLOS ONE. After careful consideration, we feel that it has merit but does not fully meet PLOS ONE’s publication criteria as it currently stands. Therefore, we invite you to submit a revised version of the manuscript that addresses the points raised during the review process.

We look forward to receiving your revised manuscript.

Kind regards,

Ji-Zhong Wan

Academic Editor

PLOS ONE

Journal Requirements:

We note that Figures 2, 3b, 4, 5 in your submission contain [map/satellite] images which may be copyrighted. All PLOS content is published under the Creative Commons Attribution License (CC BY 4.0), which means that the manuscript, images, and Supporting Information files will be freely available online, and any third party is permitted to access, download, copy, distribute, and use these materials in any way, even commercially, with proper attribution. For these reasons, we cannot publish previously copyrighted maps or satellite images created using proprietary data, such as Google software (Google Maps, Street View, and Earth). For more information, see our copyright guidelines: http://journals.plos.org/plosone/s/licenses-and-copyright.

2a, You may seek permission from the original copyright holder of Figures 2, 3b, 4, 5 to publish the content specifically under the CC BY 4.0 license. 

2b, If you are unable to obtain permission from the original copyright holder to publish these figures under the CC BY 4.0 license or if the copyright holder’s requirements are incompatible with the CC BY 4.0 license, please either i) remove the figure or ii) supply a replacement figure that complies with the CC BY 4.0 license. Please check copyright information on all replacement figures and update the figure caption with source information. If applicable, please specify in the figure caption text when a figure is similar but not identical to the original image and is therefore for illustrative purposes only.

Please include captions for your Supporting Information files at the end of your manuscript, and update any in-text citations to match accordingly. Please see our Supporting Information guidelines for more information: http://journals.plos.org/plosone/s/supporting-information.

Reviewers' comments:

Reviewer's Responses to Questions

**Comments to the Author**

1. Is the manuscript technically sound, and do the data support the conclusions?

Reviewer #1: Yes

Reviewer #2: Yes

2. Has the statistical analysis been performed appropriately and rigorously? 

Reviewer #1: Yes

Reviewer #2: Yes

3. Have the authors made all data underlying the findings in their manuscript fully available?

Reviewer #1: Yes

Reviewer #2: No

4. Is the manuscript presented in an intelligible fashion and written in standard English?

Reviewer #1: Yes

Reviewer #2: Yes

5. Review Comments to the Author

Reviewer #1: Comments

This study highlights the district level biogeographic regionalization of the Asteraceae species in the Balsas regions of Mexico. Study is very interesting and innovative highlighting different environment variables to be responsible for varied species richness and their distribution. This study has potential for identifying smaller biotic regions for endemic and other ecologically important species to conserve and can be replicated in similar regions. However, few points are of concern here and are summarized below to be addressed by the authors.

Abstract:

P2-L42: differ between them,…. Replace with ‘differ significantly’

P2-L43: than in the Upper Balsas… Shouldn’t it be ‘lower Balsas’ ? Please check

Introduction:

P3-L59: in terms of their endemic taxa….. I wonder if this regionalization meant to be specific for endemic taxa? Its always in broader terms referring to biota. Please recheck.

P4-L109: regionalization …….BD. insert ‘of’ before ‘the SDTF’.

P5-L115: The regionalization………..Replace ‘DB’ with ‘BD’.

Materials and methods:

Spatial data

P5-L143: [55]: 1) This is confusing. Please insert ‘as:’ before 1)

As per point no.1 How did you geo-reference the points with no co-ordinates? Please clear.

P6-L154: ‘Environmental variables’. Should you be consistent with the terms i.e. variables/predictors?

P7-L182-184: We used the dissimilarity…….Biodiverse program. This looks redundant with the previous paragraph (P6 L160, L165-166). Could you try to club these together and put extra information in the sentence?

Ordination analysis

P7-L195: along of reduced ….. Replace ‘of’ with ‘the’.

P7-L201-202: We extracted…… NMDS. Replace ‘ArcGis’ with ‘ArcGIS’.

P7-L202: Should you add any figure no. for refence?

Selection of SDTF environmental predictors

P7-L204: First we considered……… (S1 Table). This is confusing as I observed there are only 32 uncorrelated variables and not 58 in S1. Therefore mention the ‘S1 table’ reference after uncorrelated variables (P8-L214).

Results

P8-L234-235: The differentiated…….. species. Rephrase the sentence i.e. the differentiated groups shown in the dendrogram (Fig 3, a) represent exclusive species of the groups (S2 Table).

P8-L234: Replace ‘Fig 3’ with ‘(Fig 3, a)’.

P8-L229-230; P9-L251-257: Although……… Balsas; The biogeographic tracks…… composition. Be consistent with the use of words. i.e. in the MS mostly groups and districts are used alternatively which might be confusing for the readers with abrupt appearance in the paragraph.

P9-L253-257: Each identified……. Altitudes. Same as previous comment. The sentence becomes very confusing for the common readers because of the use of alternative terminology e.g. lower Balsas or track. Insert ‘in the upper Balsas’ after eastern track.

SDTF environmental predictors

P10-L270: The most parsimonious …….. Here 10 variables are mentioned however, in the table 1 only 9 variables are shown. Please confirm.

Discussion

P11-L297: Our results….. Rephrase the sentence i.e. Our results are in congruence with……

P11-L297: Replace ‘DB’with ‘BD’.

P11-L309: Remove ‘BD’.

P11-L312: These results…….. China. Is the study mentioned represent similar region i.e. SDTF. Also the China study highlights the richness correlation with annual precipitation however in the Balsas the dominance of herbs (58%) might be influenced by seasonal environmental variable as rightly captured in this study. You may use other reference for this.

P12-L334-335: It has been….. (Fig 4). Should you replace the “(Fig 4)” with (Fig 3)? Since there is no descriptive label to represent group 3.

Figures

Figure 3 b: Could you add labels for group ? i.e. Group 3: Upper Balsas District etc. for readers understanding.

Table

S1 table: Are there no representative species in group 2? How it was delineated as separate district based on the analysis?

Reviewer #2: This study analyzed patterns of bioregionalization for Asteraceae in the Mexican Balsas Depression. The authors found bioregions and explored environmental correlates of species turnover and richness. Overall I found the approach correct and the results interesting.

I have only a few comments.

It is not clear if SDTF is distributed only within the BD limits or also occurs outside of it (and where). If it has a wider distribution than BD, then clarify in the Introduction if the region of interest is BD or SDTF and why. Moreover, in the first objective, it is hard to tell if the unity of study is BD or SDTF.

Authors opted for the WPGMA clustering algorithm, however, UPGMA was found to have a higher performance for bioregionalization than WPGMA (Kreft & Jetz 2010 A framework for delineating biogeographical regions based on species distributions, J. Biogeography 37:2029-2053). Other potential approaches are those based on network analysis (Edler et al 2017 Infomap bioregions: Interactive mapping of biogeographical regions from species distributions. Systematic Biology, 66:197–204) or DAPC (Maestri & Duarte 2020 Evoregions: Mapping shifts in phylogenetic turnover across biogeographic regions. Methods in Ecology and Evolution 11:1652-1662).

Lines 80-81: Regionalization can find regions defined by the endemicity of very few species, and thus unlikely to serve as ‘biodiversity hotspots'.

6. PLOS authors have the option to publish the peer review history of their article (what does this mean?). If published, this will include your full peer review and any attached files.

Reviewer #1: No

Reviewer #2: No

---

## [Author Response · Author response to Decision Letter 0]

27 Apr 2021

Point-by-point Response to the Journal requirements and reviewers' comments.

Journal Requirements:

1.-Please ensure that your manuscript meets PLOS ONE's style requirements, including those for file naming. The PLOS ONE style templates can be found at

Author´s reply: The files were renamed according to the editorial standards mentioned.

 We note that Figures 2, 3b, 4, 5 in your submission contain [map/satellite] images which may be copyrighted. All PLOS content is published under the Creative Commons Attribution License (CC BY 4.0), which means that the manuscript, images, and Supporting Information files will be freely available online, and any third party is permitted to access, download, copy, distribute, and use these materials in any way, even commercially, with proper attribution. For these reasons, we cannot publish previously copyrighted maps or satellite images created using proprietary data, such as Google software (Google Maps, Street View, and Earth). For more information, see our copyright guidelines: http://journals.plos.org/plosone/s/licenses-and-copyright.

 Author´s reply: All figures were created by us and no other map with prior copyright was used, so PLOS may publish it under Creative Commons Attribution License (CC BY 4.0). For this reason, figures 2, 3b, 4 and 5 will not be removed from the shipment.

3.- Please include captions for your Supporting Information files at the end of your manuscript, and update any in-text citations to match accordingly. Please see our Supporting Information guidelines for more information: http://journals.plos.org/plosone/s/supporting-information.

Author´s reply: The subtitles of the supplementary material were added at the end of the main text as suggested by the editorial guidelines (See Lines 677-679).

Reviewer #1: Comments

This study highlights the district level biogeographic regionalization of the Asteraceae species in the Balsas regions of Mexico. Study is very interesting and innovative highlighting different environment variables to be responsible for varied species richness and their distribution. This study has potential for identifying smaller biotic regions for endemic and other ecologically important species to conserve and can be replicated in similar regions. However, few points are of concern here and are summarized below to be addressed by the authors.

Abstract:

P2-L42: differ between them,…. Replace with ‘differ significantly’

Author´s reply: The authors thanks for you recomendation.

P2-L43: than in the Upper Balsas… Shouldn’t it be ‘lower Balsas’ ? Please check

Author´s reply: We appreciate the observation. it is indeed ‘Lower Balsas’

Introduction:

P3-L59: in terms of their endemic taxa….. I wonder if this regionalization meant to be specific for endemic taxa? Its always in broader terms referring to biota. Please recheck.

Author´s reply: We change “taxa” for “biota” in the text, which would be the most appropriate term.

P4-L111: regionalization …….BD. insert ‘of’ before ‘the SDTF’.

Author´s reply: done.

P5-L117: The regionalization………..Replace ‘DB’ with ‘BD’.

Author´s reply: We appreciate your observation.

Materials and methods:

Spatial data

P5-L145: [55]: 1) This is confusing. Please insert ‘as:’ before 1)

As per point no.1 How did you geo-reference the points with no co-ordinates? Please clear.

Author´s reply: We add the georeferencing explained in the following lines 145-146.

P6-L157: ‘Environmental variables’. Should you be consistent with the terms i.e. variables/predictors?

Author´s reply: We add the observation. We change the term ‘environmental variables’ for ‘environmental predictors’.

P7-L185-190: We used the dissimilarity…….Biodiverse program. This looks redundant with the previous paragraph (P6 L160, L168-170). Could you try to club these together and put extra information in the sentence?

Author´s reply: We rewrite the first line of P7 (Line 185), considering the reviewer's suggestion. Remaining as follows: 

The dissimilarity matrix was used (βSim) for cluster analysis,…

Ordination analysis

P7-L198: along of reduced ….. Replace ‘of’ with ‘the’.

Author´s reply: done. We appreciate your observation.

P7-L204-205: We extracted…… NMDS. Replace ‘ArcGis’ with ‘ArcGIS’.

Author´s reply: done. We appreciate your observation.

P7-L205: Should you add any figure no. for refence?

Author´s reply: The figure showing the results of this part of the method is cited in the results and corresponds to Figure 5.

Selection of SDTF environmental predictors

P7-L207: First we considered……… (S1 Table). This is confusing as I observed there are only 32 uncorrelated variables and not 58 in S1. Therefore mention the ‘S1 table’ reference after uncorrelated variables (P8-L217).

Author´s reply: We appreciate the observation. This was addressed as suggested by the reviewer.

Results

P8-L237-238: The differentiated…….. species. Rephrase the sentence i.e. the differentiated groups shown in the dendrogram (Fig 3, a) represent exclusive species of the groups (S2 Table).

Author´s reply: The dendrogram shows the grouping of the 571 species used in this study, after the grouping, the exclusive species of each group were identified, which are listed in S2 Table. We modify the wording of the paragraph.

The differentiated groups shown in the dendrogram (Fig. 3, a) are represented by species exclusive to these groups (Table S2).

P8-L237: Replace ‘Fig 3’ with ‘(Fig 3, a)’.

Author´s reply: done. We appreciate your observation.

P8-L232-238; P9-L258-262: Although……… Balsas; The biogeographic tracks…… composition. Be consistent with the use of words. i.e. in the MS mostly groups and districts are used alternatively which might be confusing for the readers with abrupt appearance in the paragraph.

Author´s reply: We homologated the terms that were used as synonyms and the rest were defined the first time they were used as in the case of districts and tracks. See L245-247 and L260-262.

P9-L258-262: Each identified……. Altitudes. Same as previous comment. The sentence becomes very confusing for the common readers because of the use of alternative terminology e.g. lower Balsas or track. Insert ‘in the upper Balsas’ after eastern track.

SDTF environmental predictors

Author´s reply: We add more information in this paragraph to make it more understandable.

P10-L275: The most parsimonious …….. Here 10 variables are mentioned however, in the table 1 only 9 variables are shown. Please confirm.

Author´s reply: We appreciate the observation. It has been corrected.

Discussion

P11-L302: Our results….. Rephrase the sentence i.e. Our results are in congruence with……

Author´s reply: We appreciate the suggestion.

P11-L302: Replace ‘DB’with ‘BD’.

Author´s reply: done. We appreciate the observation.

P11-L315: Remove ‘BD’.

Author´s reply: done.

P11-L318: These results…….. China. Is the study mentioned represent similar region i.e. SDTF. Also the China study highlights the richness correlation with annual precipitation however in the Balsas the dominance of herbs (58%) might be influenced by seasonal environmental variable as rightly captured in this study. You may use other reference for this.

Author´s reply: The study by Zhang et al. (2016) was carried out in an SDTF. Regarding the ratio of the seasonality of precipitation and dominance of herbs it is addressed in the following paragraph. In our search, we did not find a study carried out in the SDTF with which they found a positive relationship of the richness of herb species with the seasonality of the precipitation.

P12-L340-341: It has been….. (Fig 4). Should you replace the “(Fig 4)” with (Fig 3)? Since there is no descriptive label to represent group 3.

Author´s reply: done. We thank you for the suggestion.

Figures

Figure 3 b: Could you add labels for group ? i.e. Group 3: Upper Balsas District etc. for readers understanding.

Author´s reply: The Figure 3b was modified considering the recommendations of the reviewer.

Table

S1 table: Are there no representative species in group 2? How it was delineated as separate district based on the analysis?

Author´s reply: The districts were established after the consensus, that is, when the unrepresentative groups (groups: 1,2,5,6,7,8) were reassigned according to their floristic similarity to groups 3 and 4. See Lines 191-194.

Reviewer #2: 

This study analyzed patterns of bioregionalization for Asteraceae in the Mexican Balsas Depression. The authors found bioregions and explored environmental correlates of species turnover and richness. Overall I found the approach correct and the results interesting.

I have only a few comments.

It is not clear if SDTF is distributed only within the BD limits or also occurs outside of it (and where). If it has a wider distribution than BD, then clarify in the Introduction if the region of interest is BD or SDTF and why. Moreover, in the first objective, it is hard to tell if the unity of study is BD or SDTF.

Author´s reply: We include information about the distribution of the SDTF in Lines 95-97. We added information to clarify that the main area of interest was the surface occupied by the SDTF within the BD (L108). Finally, objective one was rewritten to clarify that the unit of study was the SDTF within the BD.

Authors opted for the WPGMA clustering algorithm, however, UPGMA was found to have a higher performance for bioregionalization than WPGMA (Kreft & Jetz 2010 A framework for delineating biogeographical regions based on species distributions, J. Biogeography 37:2029-2053). Other potential approaches are those based on network analysis (Edler et al 2017 Infomap bioregions: Interactive mapping of biogeographical regions from species distributions. Systematic Biology, 66:197–204) or DAPC (Maestri & Duarte 2020 Evoregions: Mapping shifts in phylogenetic turnover across biogeographic regions. Methods in Ecology and Evolution 11:1652-1662).

Author´s reply: In this case, we consider that the weighting of the contribution of the clusters by the number of terminal nodes of the WPGMA method favors our results due to the discrepancy in the number of taxa in each cell, ensuring that each cell contributes the same way to the cluster. to which it belongs. In addition, the performance of the WPGMA is considered as successful as the UPGMA (Kreft and Jetz, 2010).

Lines 80-81: Regionalization can find regions defined by the endemicity of very few species, and thus unlikely to serve as ‘biodiversity hotspots'.

Author´s reply: In this same paragraph we argue why a bioregion can act as a biodiversity hotspot. In response to the reviewer's comment, regions may be defined by few species, but these taxa may be rare, endemic, or in some critical state. By identifying these important areas and communities, this information can help design reserves that can protect the biodiversity more efficiently.

---

## [Decision Letter · Decision Letter 1]

31 May 2021

Biogeographic regionalization by spatial and environmental components: a numerical proposal

PONE-D-21-01262R1

Dear Dr. Flores,

We’re pleased to inform you that your manuscript has been judged scientifically suitable for publication and will be formally accepted for publication once it meets all outstanding technical requirements.

Kind regards,

Ji-Zhong Wan

Academic Editor

PLOS ONE

Additional Editor Comments (optional):

Reviewers' comments:

Reviewer's Responses to Questions

**Comments to the Author**

1. If the authors have adequately addressed your comments raised in a previous round of review and you feel that this manuscript is now acceptable for publication, you may indicate that here to bypass the “Comments to the Author” section, enter your conflict of interest statement in the “Confidential to Editor” section, and submit your "Accept" recommendation.

Reviewer #2: All comments have been addressed

2. Is the manuscript technically sound, and do the data support the conclusions?

Reviewer #2: Yes

3. Has the statistical analysis been performed appropriately and rigorously? 

Reviewer #2: Yes

4. Have the authors made all data underlying the findings in their manuscript fully available?

Reviewer #2: (No Response)

5. Is the manuscript presented in an intelligible fashion and written in standard English?

Reviewer #2: Yes

6. Review Comments to the Author

Reviewer #2: (No Response)

7. PLOS authors have the option to publish the peer review history of their article (what does this mean?). If published, this will include your full peer review and any attached files.

Reviewer #2: No

---

## [Editor Report · Acceptance letter]

7 Jun 2021

PONE-D-21-01262R1 

Biogeographic regionalization by spatial and environmental components: numerical proposal 

Dear Dr. Flores-Tolentino:

I'm pleased to inform you that your manuscript has been deemed suitable for publication in PLOS ONE. Congratulations! Your manuscript is now with our production department. 

Kind regards, 

on behalf of

Dr. Ji-Zhong Wan 

Academic Editor

PLOS ONE